# Effects of siRNA-Mediated Knockdown of GSK3β on Retinal Ganglion Cell Survival and Neurite/Axon Growth

**DOI:** 10.3390/cells8090956

**Published:** 2019-08-22

**Authors:** Zubair Ahmed, Peter J. Morgan-Warren, Martin Berry, Robert A. H. Scott, Ann Logan

**Affiliations:** 1Neuroscience and Ophthalmology, Institute of Inflammation and Ageing, University of Birmingham, Birmingham B15 2TT, UK; 2Academic Department of Military Surgery and Trauma, Royal Centre for Defence Medicine, Birmingham B45 9NU, UK; 3Birmingham and Midland Eye Centre, Birmingham B18 7QH, UK

**Keywords:** retinal ganglion cells, GSK3β, RTP801, neurite outgrowth, optic nerve injury, axon regeneration

## Abstract

There are contradictory reports on the role of the serine/threonine kinase isoform glycogen synthase kinase-3β (GSK3β) after injury to the central nervous system (CNS). Some report that GSK3 activity promotes axonal growth or myelin disinhibition, whilst others report that GSK3 activity prevents axon regeneration. In this study, we sought to clarify if suppression of GSK3β alone and in combination with the cellular-stress-induced factor RTP801 (also known as REDD1: regulated in development and DNA damage response protein), using translationally relevant siRNAs, promotes retinal ganglion cell (RGC) survival and neurite outgrowth/axon regeneration. Adult mixed retinal cell cultures, prepared from rats at five days after optic nerve crush (ONC) to activate retinal glia, were treated with siRNA to GSK3β (siGSK3β) alone or in combination with siRTP801 and RGC survival and neurite outgrowth were quantified in the presence and absence of Rapamycin or inhibitory Nogo-A peptides. In in vivo experiments, either siGSK3β alone or in combination with siRTP801 were intravitreally injected every eight days after ONC and RGC survival and axon regeneration was assessed at 24 days. Optimal doses of siGSK3β alone promoted significant RGC survival, increasing the number of RGC with neurites without affecting neurite length, an effect that was sensitive to Rapamycin. In addition, knockdown of GSK3β overcame Nogo-A-mediated neurite growth inhibition. Knockdown of GSK3β after ONC in vivo enhanced RGC survival but not axon number or length, without potentiating glial activation. Knockdown of RTP801 increased both RGC survival and axon regeneration, whilst the combined knockdown of GSK3β and RTP801 significantly increased RGC survival, neurite outgrowth, and axon regeneration over and above that observed for siGSK3β or siRTP801 alone. These results suggest that GSK3β suppression promotes RGC survival and axon initiation whilst, when in combination with RTP801, it also enhanced disinhibited axon elongation.

## 1. Introduction

The role of the serine/threonine kinase isoform glycogen synthase kinase-3β (GSK3β) [1,2,3,4], which is expressed in the central nervous system (CNS) [5], is controversial, since GSK3 activity is reported by some to promote axonal growth or myelin disinhibition, whilst other studies claim the opposite is true [6,7,8,9,10]. These reported differences have been attributed to a variety of factors including cell type, neuronal age, axon environment, and the ability of GSK3 to influence the activity of multiple downstream targets and, hence, the adoption of several diverse functions in nerve regeneration [11]. For example, collapsin response mediator protein 2 (CRMP2) and microtubule-associated protein 1B are both substrates for GSK3, but CRMP2, which is inhibited by GSK3-mediated phosphorylation, promotes microtubule polymerisation and myelin disinhibition, whilst MAP1B, which is directly phosphorylated by GSK3-mediated phosphorylation, promotes axon growth [9,12,13,14,15,16]. It has also been suggested that ectopic expression of non-physiological levels of GSK3 may be detrimental, since the focus on one of the two isoforms of GSK3, mainly GSK3β, may elicit compensatory responses by GSK3α and vice-versa [10].

In an attempt to clear up some of the potential pitfalls mentioned above, Gobrecht et al. (2014) [9] used well-defined phosphorylation resistant GSK3α^S21A^/β^S9A^ [GSK3(α/β)^S/A^] double knock-in mice to show that elevated GSK3 activity accelerated peripheral nerve regeneration [9]. This observed effect was based on phosphorylated MAP1B-associated inhibition of microtubule detyrosination and subsequent increase in microtubule dynamics in axon growth cones [9,17]. However, using the transgenic as well as the conditional retinal ganglion cell (RGC) specific GSK3α and GSK3β knock out (KO) mice, the elevation of GSK3 activity in GSK3^S/A^ mice was compromised, and GSK3β KO potentiated inflammatory stimulation-mediated RGC axon regeneration [10]. These effects were associated with varying degrees of inactive CRMP2 in optic nerve axons, whereas no CRMP2 phosphorylation was observed in peripheral nerve regeneration [10]. Furthermore, elevated GSK3 activity enhanced neurite outgrowth in RGC only when constant CRMP2 activity was maintained, suggesting that CRMP2 inhibition masks the positive effects of MAP1B activity in CNS neurons [10].

Whilst Rapamycin inhibition of the mammalian target of Rapamycin (mTOR) has no effect on GSK3β activation, the inhibition of GSK3β reportedly activates mTOR in cultured dorsal root ganglion neurons (DRGN) [3,18], presumably by releasing the tuberous sclerosis complex (TSC) from direct GSK3β-dependent activation [19] (Figure 1). Thus, suppression of GSK3β activity may enhance CNS neuron (including RGC) survival and disinhibit axon growth. We recently demonstrated that siRNA-mediated suppression of the cellular-stress-induced factor RTP801 (also known as REDD1, or DDIT4-regulated in development and DNA damage response protein or DDIT4–DNA damage inducible transcript 4), an approach that is already translated to the clinic, promotes RGC survival and limited axon elongation in vivo after optic nerve crush (ONC) [20]. We suggest that, since siRTP801-induced mTORC1/2 activity may partly be suppressed by GSK3β, additional inhibition of GSK3β may potentiate siRTP801-induced mTORC1/2 activity and consequent RGC survival and axon regeneration effects by reducing TSC. Accordingly, a siGSK3β + siRTP801 combinatorial treatment may enhance RGC neuroprotective/axogenic signalling after ONC. Here, we report the results of in vitro and in vivo experiments, investigating the effects of single siGSK3β and dual siGSK3β + siRTP801 treatment on downstream substrate activity in RGC in the presence of activated retinal glia.

## 2. Materials and Methods

### 2.1. Reagents

All reagents were purchased from Sigma (Poole, UK) unless otherwise stated.

### 2.2. Experimental Design

In vitro and in vivo experiments were designed to evaluate a role for GSK3β in RGC neuroprotection and axon regeneration.

For in vitro experiments, retinal cultures were used to evaluate the effects of siGSK3β and/or siRTP801 using treatment regimens and groups, detailed in Figure 2A, by investigators masked to treatment conditions. Dissociated mixed adult rat retinal cultures were prepared from injured rats at 5 days after ONC to activate the retinal glia, as described previously [21]. Briefly, retinal cells were cultured on poly-D-lysine and laminin-coated glass slides and transfected using Lipofectamine 2000 reagent (Invitrogen, Paisley, UK) with either 50 nM siGSK3β, 25 nM siGSK3β + 25 nM siRTP801, 50 nM siEGFP, or culture medium alone, and incubated for 3 days in supplemented Neurobasal-A (sNBA; containing supplement B and l-glutamine; all from Invitrogen). Cultures were then fixed in 4% paraformaldehyde and immunostained with a variety of markers for analysis of RGC survival, neurite outgrowth, siGSK3β/siRTP801-target knockdown, and reactive glial marker expression. Specific wells were also treated with rapamycin (10 nM) and added at the end of the siRNA transfection period [21]. In vitro experiments composed of two to three wells per treatment condition, repeated with retinae from at least three independent animals. 

For in vivo experiments, adult male Sprague-Dawley rats (6–8-week-old, 180–220 g; Charles River, Margate, UK) were randomly assigned to treatment groups, and investigators were masked to treatment conditions during analyses. In experiments to evaluate the contribution of GSK3β to RGC survival and axon regeneration, 20 µg of siGSK3β was intravitreally injected in one eye and 20 µg control siRNA to enhanced green fluorescent protein (siEGFP) was injected into the fellow eye on days 0, 8, and 16 (Figure 2B). In further experiments, and to evaluate the contribution of GSK3β to RTP801-mediated activation of the mTOR pathway in RGC neuroprotection and axon regeneration, either 20 µg siGSK3β + 20 µg siEGFP (control siEGFP was used to complement total siRNA dose) or 20 µg siGSK3β + 20 µg siRTP801 was intravitreally injected into one eye with 40 µg siEGFP into the fellow eye on days 0, 8, and 16. In further experiments, and to evaluate the contribution of GSK3β to RTP801-mediated activation of the mTOR pathway in RGC neuroprotection and axon regeneration, either 20 µg siGSK3β + 20 µg siEGFP (control siEGFP was used to complement total siRNA dose) or 20 µg siGSK3β + 20 µg siRTP801 was intravitreally injected into one eye with 40 µg siEGFP into the fellow eye on days 0, 8, and 16. Tissues were harvested at day 24 for immunohistochemistry and analysis of RGC survival, axon regeneration, glial activation, and GSK3β/RTP801 expression. Uninjured eyes were used as intact controls, acting as a benchmark for determining the effects of siGSK3β and siGSK3β + siRTP801 compared to siEGFP controls after ONC.

### 2.3. Small Interfering RNA (siRNA)

All siRNA compounds used in this study were chemically synthesized at BioSpring GmbH (Frankfurt, Germany) for Quark Pharmaceuticals Inc. (Newark, CA, USA), and provided to us as a gift. siEGFP and siGSK3β compounds were as previously described [21]. The sequence and chemical modifications to siRTP801 are proprietary to Quark Pharmaceuticals Inc., and the siRNAs may be obtained on request.

### 2.4. In Vitro Experiments

#### 2.4.1. Adult Rat Retinal Cultures

To assess the effects of siGSK3β with and without siRTP801 on RGC survival and neurite outgrowth in the presence of activated glia, mimicking the in vivo injury-induced scenario, retinal cell cultures were prepared from animals 5 days after ONC [21]. Briefly, adult male Sprague-Dawley rats were killed by CO_2_ overdose and retinae was harvested and dissociated into single cell suspensions using a papain dissociation kit, following the manufacturer’s instructions (Worthington Biochemicals, Lakewood, NJ, USA) [21]. Retinal cells were plated at a density of 125,000 cells per well in 8-well chamber slides (BD Biosciences, Watford, UK), pre-coated with poly-d-lysine and laminin in sNBA (Invitrogen), and were incubated at 37 °C and 5% CO_2_. Cultures were allowed to settle overnight before treatment the next day. The identity of wells was masked from the investigator and experiments were performed in duplicate and repeated on 3 independent occasions.

#### 2.4.2. Knockdown of GSK3β and GSK3β + RTP801 in Retinal Cultures

Retinal cells were transfected with Lipofectamine 2000, as previously described [21]. Briefly, the siRNA and Lipofectamine 2000 were diluted in sNBA, mixed gently to form complexes, and added to cells dropwise for 5 h before addition of sNBA to a final volume of 500 µL/well and incubated at 37 °C and 5% CO_2_ for 3 days. Culture medium alone and Lipofectamine 2000 served as controls. Some wells were also treated with Rapamycin at 10 nM and added at the end of the siRNA transfection period [21] to determine the effects of mTOR signalling in siGSK3β and/or siRTP801-mediated RGC survival and neurite outgrowth. Experiments were performed in duplicate wells and repeated on 3 independent occasions.

A dose–response assay was undertaken to identify the optimum concentration of siRNA for use in later experiments using the following treatment conditions: sNBA culture medium alone; Lipofectamine transfection reagent; siEGFP (at 10 nM, 20n M, 50 nM, 100 nM, and 200 nM); and siGSK3β (at 10 nM, 20 nM, 50 nM, 100 nM, and 200 nM), with up to 3 wells/condition (repeated on 3 independent occasions). High concentrations (>100 nM) of siRNA doses were toxic to RGC and reduced their viability. Cultures were incubated for 3 days and subsequently analysed for βIII-tubulin^+^ RGC survival, neurite outgrowth initiation/elongation, and immunohistochemical localisation of GSK3β in RGC, as described below.

For the combined siRTP801 + siGSK3β experiments, retinal cell cultures were established from rats 5 days after ONC, as described above, grown in sNBA culture medium, and transfected using Lipofectamine 2000 with 50 nM siGSK3β, 50 nM siRTP801, or 25 nM siGSK3β + 25 nM siRTP801 and 50 nM siEGFP (to give an equivalent total siRNA concentration to the test combination). RGC survival, neurite outgrowth, and RGC localisation of RTP801 and GSK3β were evaluated by immunocytochemistry, as described below. In these experiments, up to 3 wells/condition were used and each was repeated on 3 independent occasions.

#### 2.4.3. Confirmation of siRNA Specificity and Target Knock-Down with qPCR

RNA was obtained from mixed retinal cultures incubated under appropriate experimental conditions, and with siRNA, as described above. The specificity of the GSK3β primer was confirmed by running a qPCR reaction with cDNA from an untreated sample in duplicate and undertaking melt–curve analysis to detect a single amplicon [21]. GSK3β and RTP801 knockdown was evaluated by measuring fold-changes in GSK3β and RTP801 mRNA expression, relative to the housekeeping gene GAPDH, in samples from the different treatment conditions, as described below. Experiments were performed in duplicate and repeated on 3 independent occasions.

#### 2.4.4. Immunocytochemistry

After incubation for 3 days, cell culture medium was removed from wells and fixed in 4% PFA in PBS for 10 min before immunostaining, as described previously [21]. Briefly, cells were washed three times in rinsing buffer (0.1% Triton X-100 in PBS) and nonspecific protein binding was blocked with 10% normal goat/donkey serum and 3% BSA in PBS for 1 h at room temperature. Primary antibodies diluted in antibody diluting buffer (3% bovine serum albumin and 0.05 Tween-20 in PBS) (Table 1) were added to wells and incubated for 1 h at RT before washing cells in rinsing buffer and incubation with Alexa 488 or Alexa 594-labelled secondary antibodies (Table 1). After further washes, coverslips were mounted with Vectashield containing DAPI (Vector Laboratories, Peterborough, UK) and viewed under a Zeiss Axioplan 2 epi-fluorescence microscope, equipped with an AxioCam HRc and running Axiovision Software (all from Carl Zeiss Ltd., Hertfordshire, UK).

#### 2.4.5. Assessment of RGC Survival and Neurite Outgrowth in Retinal Cultures

RGC survival and neurite outgrowth in the retinal cultures were quantified from captured images, as described previously [21]. Briefly, each well was divided into 9 squares and 4 photomicrographs were captured within each square, giving a total of 36 images/well. Surviving βIII-tubulin^+^ RGCs were counted from each of these images and RGC numbers/well were determined. The number of βIII-tubulin^+^ RGCs extending neurite(s) over a length greater than the diameter of the somata were also counted, as well as the length of the longest neurite/RGCs measured using Image Pro 6.2 (Media Cybernetics, Bethesda, MD, USA). The proportion of siGSK3β^+^/βIII-tubulin^+^ RGCs were expressed as a % of total numbers of βIII-tubulin^+^ RGCs. The number of GFAP^+^ astrocytes was also counted in a similar way to RGCs.

#### 2.4.6. Disinhibition of Neurite Outgrowth on Nogo-P4 Inhibitory Peptide after siGSK3β-Mediated Knockdown

Glial activated retinal cultures were prepared 5 days after ONC, as described above, and treated with recombinant Nogo-P4 peptide (Alpha Diagnostics, San Antonio, TX, USA) at a final concentration of 25 µM known to inhibit RGC neurite outgrowth (pre-determined in a separate dose response assay) in a final volume of 500 µL/well. The Nogo-P4 peptide was added to cultures after the 5 h period for siRNA transfection and left for the full duration of the experiment. Controls comprised cultures of sNBA culture medium alone and cultures with added Lipofectamine 2000 + ciliary neurotrophic factor (CNTF; 20 ng/mL) as a positive control [22]. Experiments were undertaken using up to 3 wells/condition and were repeated on 3 independent occasions using cultures of retinal cells obtained from the eyes of different rats.

### 2.5. In Vivo Experiments

#### 2.5.1. Animals and Surgical Procedures

Adult male Sprague Dawley rats (Charles River, Kent, UK), weighing 200–250 g at the commencement of experiments, were housed at 21 °C/55% humidity in a 12 h-light-12 h-dark cycle with ad libitum access to food and water. All surgery was carried out in accordance with the UK home office regulations for the care and use of experimental animals and the UK Animals (Scientific Procedures) Act 1986, licensed by the UK home office and approved by the University of Birmingham animal welfare and ethical review board. Experiments with animals also conformed to the association for research into vision and ophthalmology (ARVO) statement for the use of animals in research, except the bilateral ONC, which is imposed by the UK home office and is considered as ‘reduction’ in keeping with the 3 R’s principles.

Optic nerves were crushed bilaterally in anesthetized rats using calibrated watchmaker’s forceps through a supra-orbital approach 2 mm from the lamina cribrosa, preserving the dura and retinal vascular supply, as described previously [21]. Preliminary dose–response studies showed that 20 µg of siGSK3β injected immediately after ONC optimally reduced GSK3β mRNA in the retina. Intravitreal injections were performed using glass micropipettes prepared in-house. Animals were injected with 20 µg siRNA, dissolved in 10 µL PBS, with intravitreal delivery of siGSK3β in one eye (*n* = 6) and control siEGFP to the contralateral eye (*n* = 6), and this was repeated on day 8 and 16 after ONC, as described previously [21,23]. Tissues were harvested at 24 days after ONC and processed for immunohistochemistry, as described below. An additional group of uninjured animals were similarly processed and used as controls (*n* = 6 eyes).

To evaluate the effects of siGSK3β + siRTP801, ONC was performed, as described above, in 12 rats (24 eyes) and immediately injected intravitreally with 10 µL PBS containing 40 µg siEGFP (*n* = 6 eyes), 20 µg siGSK3β + 20 µg siEGFP (siEGFP used to make up equivalent dose of combined siRNAs; *n* = 6 eyes), and 20 µg siGSK3β + 20 µg siRTP801 (*n* = 6 eyes). Intravitreal injections of these treatments were repeated at 8 days and 16 days post-ONC. Rats receiving intravitreal PBS without ONC were used as baseline controls (*n* = 6 eyes). Tissues were harvested at 24 days after ONC and processed for immunohistochemistry, as described below.

#### 2.5.2. Tissue Preparation

Animals were killed 24 days after ONC by rising concentrations of CO_2_ and processed for immunohistochemistry as described previously [21]. Briefly, animals were transcardially perfused with PBS followed by 4% paraformaldehyde (PFA; TAAB, Reading, UK). Eyes and ON were post-fixed for a further 2 h before cryoprotection through a graded series of sucrose solutions, blocked up in optimal cutting temperature embedding medium (OCT; Thermo Fisher, Runcorn, UK), and 15 µm thick radial sections of eyes and longitudinal sections of ON were cut on a Bright cryostat (Brights Instrument, Huntingdon, UK), adhered onto Superfrost Plus microscope slides (Fischer Scientific, Loughborough, UK) and stored at −20 °C until required. Sections of the ON containing a defined lesion site and radial ocular sections taken through the ON head were selected for further analysis.

#### 2.5.3. Immunohistochemistry

Immunohistochemistry for a variety of markers were performed, as described previously [21]. Briefly, eye and ON sections were thawed at room temperature, permeabilised in rinsing buffer containing 0.1% Triton X-100 in PBS and blocked in 10% normal goat serum/donkey serum and 3% bovine serum albumin (BSA) for 1 h in a humidified chamber. Sections were then incubated with primary antibodies (Table 1), diluted in antibody diluting buffer (ADB) containing 3% BSA in PBS overnight at 4 °C in a humidified chamber. Sections were then washed in several changes of PBS and incubated with fluorescently-labelled secondary antibodies (Alexa 488 or Alexa 594; Invitrogen), diluted in ADB, and incubated for 1 h at RT. Sections were then washed in PBS and coverslips were mounted using Vectashield containing DAPI (Vector Laboratories) and stored in the dark at 4 °C until required for microscopic analysis, as described below.

#### 2.5.4. Assessment of RGC Survival and Axon Regeneration

All sections were masked to the treatment conditions by an independent researcher and were viewed under an upright Axioplan-2 fluorescence microscope, and images were captured using an AxioCam HRc controlled by Axiovision software (all from Carl Zeiss Ltd.). RGC survival was assessed, as previously described [21]. Briefly, RGC were counted in a standard 250 µm linear strip of the ganglion cell layer (GCL) in radial sections on either side of the ON head (four radial sections/retina, n = 6 eyes/treatment group), using the RGC antibody marker Brn3a [21] and results were expressed as the number of RGC/250 µm GCL. This method of counting RGC yields equivalent scores to those obtained from Fluorogold back-labelled or Brn3a stained retinal wholemounts [24].

Quantification of axon regeneration in longitudinal ON sections was performed, as described previously [25]. Briefly, composite images were constructed from individual ON sections, identifying the ONC site by laminin^+^ immunostaining and the number of GAP43^+^ regenerating axons extending 100, 200, 400, 800, and 1200 µm from the centre of the ONC site were counted (3 sections/ON, n = 6 eyes/group). GAP43 is the gold standard method of quantifying RGC axon regeneration in the distal segment of the rat ON and correlates with the number of axons detected by the anterograde tracer Rhodamine B [26]. The cross-sectional width of the ON was measured where axon counts were taken and used to calculate the number of axons/mm ON width to derive ∑ad, the total number of axons extending distance *d* in an ON with radius *r* using the following formula:∑ad = πr^2^ × (average number of axons/mm width)/(section thickness 0:015 mm)(1)

#### 2.5.5. Assessment of Retinal Glial Activation

Astrocytes in the GCL/nerve fibre layer (NFL) and Muller glial processes and somata were detected using GFAP and S100b immunohistochemistry and quantified as described previously [21,25]. Briefly, the number of GFAP^+^ Muller cell processes intersecting a 250 µm horizontal linear sampling line passing through the inner plexiform layer (IPL) was counted at the midpoint between the GCL and inner nuclear layer (INL) (4 sections/retina, *n* = 6 eyes/group).

### 2.6. Statistical Analysis

Data are expressed as mean ± S.E. One-way analysis of variance (ANOVA) was used to compare means, and Tukey’s post hoc test was used where appropriate. A *p*-value of less than 0.5 was considered statistically significant. All data were analysed using SPSS software (Version 20, IBM, New York, NY, USA).

## 3. Results

### 3.1. GSK3β-Induced Knockdown and Neuroprotection of RGC

Firstly, we confirmed that retinal cultures prepared from rats five days after ONC contained activated GFAP^+^ astrocytes and Müller glia, in contrast to cultures prepared from intact rats that contain few, if any, activated retinal glia (Figure 3; [21]). We then determined the optimal concentration of siGSK3β to knock down GSK3β mRNA and protein in retinal cultures and determined its effects of RGC survival and neurite outgrowth. In sNBA- and siEGFP-treated control cultures, abundant GSK3β^+^ RGCs were present. However, treatment of cultures with increasing concentrations of siGSK3β from 10–50 nM significantly reduced the numbers of GSK3β^+^ RGCs (*p* < 0.0001) to an optimum 50% compared with both sNBA-treated or siEGFP-treated control cultures (Figure 4A,B). Higher concentrations of siGSK3β (100–200 nM) were not as effective at reducing GSK3β^+^ RGCs. At the optimum concentration of siGSK3β, few GSK3β^+^/βIII-tubulin^+^ RGC were present compared to those in cultures treated with either an equimolar concentration of siEGFP or sNBA (Figure 4B). Knockdown of GSK3β mRNA with 50 nM siGSK3β was also confirmed in retinal cultures by qPCR, reducing GSK3β mRNA by 64% (Figure 4C). These results showed that siGSK3β significantly reduced GSK3β mRNA and protein and that 50 nM concentrations gave optimal knockdown in adult rat retinal cultures. Treatment of cultures with increasing concentrations of siGSK3β from 10–50 nM significantly enhanced the number of surviving βIII-tubulin^+^ RGCs (*p* < 0.0001) from 400 ± 39 to 620 ± 45 and 580 ± 37 in sNBA and 10 and 20 nM siGSK3β, respectively, reaching a maximum of 780 ± 27 RGCs after treatment with 50 nM siGSK3β (Figure 4D). This equated to almost a 2-fold increase in RGC neuroprotection, compared to either sNBA or siEGFP-treated cultures. Higher concentrations of siGSK3β (100 and 200 nM) reduced the number of RGCs and were thus potentially neurotoxic (Figure 4D). Increasing concentrations of siGSK3β up to 50 mM also significantly increased the numbers of RGC bearing neurites (*p* < 0.001) from 15 ± 1.3 to 35 ± 2% with 50 nM siEGFP or siGSK3β, respectively, equating to a 43% increase in the number of RGCs bearing neurites. Increasing the concentration of siGSK3β beyond 50 nM was detrimental to neurite outgrowth (Figure 4E). High concentrations of siEGFP were also detrimental to RGC survival. Concentration-dependent cellular toxicity has been well described in the literature and is attributed, in part, to saturation of the RNAi machinery [27]. Interestingly, increasing the concentration of siGSK3β did not affect the length of the longest neurite (Figure 4F) compared to siEGFP-treated control cultures. These results suggest that knockdown of GSK3β is RGC neuroprotective, and promotes RGC neurite growth initiation, but does not affect neurite length.

### 3.2. mTORC1-Mediation of GSK3β Effects on RGC Survival and Neurite Outgrowth

We then determined if GSK3β-mediated RGC survival and neurite outgrowth were mediated through mTORC1 activation using Rapamycin, which effectively blocks mTOR activation. Retinal cultures stained with phosphorylated (p)S6 (pS6) antibodies after treatment with either sNBA or siEGFP confirmed many pS6^+^/βIII-tubulin^+^ RGCs, along with other cells that were pS6^+^/βIII-tubulin^−^ (Figure 5A). There was no significant difference in the number of pS6^+^/βIII-tubulin^+^ RGCs between wells incubated with sNBA alone (15.8 ± 2.8%) and after transfection with siEGFP (14.3 ± 1.0%), or siGSK3β (16.4 ± 1.9%; Figure 5B). Exposure to Rapamycin abolished the expression of pS6 in both βIII-tubulin^+^ RGCs and βIII-tubulin^−^ cells (Figure 5A,B), irrespective of all other treatments applied. RGC survival was not significantly reduced in control wells by Rapamycin treatment, but Rapamycin did significantly reduce siGSK3β-enhanced RGC survival from 550 ± 27 to 400 ± 36 RGC (*p* < 0.001; Figure 5C). Neither the mean number of surviving βIII-tubulin^+^ RGC (Figure 5C), the proportion of siGSK3β-stimulated RGC bearing neurites (Figure 5D), nor the mean length of the longest neurite (Figure 5E) were affected by Rapamycin treatment. Furthermore, siGSK3β did not increase the numbers of pS6^+^ RGCs in retinal cultures (Figure 5B). These results suggest that RGC neurite outgrowth initiation was independent of mTORC1 activity, that GSK3β signalling did not regulate mTORC1 activity and that siGSK3β mediated RGC neurite outgrowth initiation occurred through mTORC1-independent pathways.

### 3.3. Treatment of Retinal Cultures with siGSK3β Disinhibits RGC Neurite Outgrowth in the Presence of Nogo Peptide

We next determined if GSK3β-mediated knockdown alone and in combination with siRTP801 can overcome the potent axon growth inhibitor, Nogo-A. The number of RGCs bearing neurites in control wells treated with sNBA and without Nogo-P4 peptide (25 ± 2%) was reduced by 50% to 12 ± 2% in the presence of inhibitory Nogo-P4 peptide (Figure 6A,B). Nogo-P4 peptide also reduced the mean neurite length in control sNBA treated cultures from 58 ± 5% to 28 ± 2% (Figure 6C). Neither Lipofectamine 2000 nor siEGFP control treatments disinhibited neurite outgrowth in the presence of Nogo-P4 peptide (Figure 6A–C). However, siGSK3β, siGSK3β + siRTP801, and CNTF similarly disinhibited RGC neurite outgrowth, increasing the proportion of RGCs bearing neurites to 30 ± 2%, 35 ± 3%, and 40 ± 3%, respectively (Figure 6A,B), and the mean neurite length to 60 ± 7 µm, 70 ± 5%, and 75 ± 5 µm, respectively (Figure 6C). Taken together, these data show that siGSK3β alone and in combination with siRTP801, stimulated RGC neurite initiation and disinhibited neurite elongation in the presence of inhibitory Nogo-P4 peptide.

### 3.4. Cellular Localisation of GSK3β and the Effects of siGSK3β on RGC Survival and Axon Regeneration 24 Days after ONC

We next determined the cellular localisation of GSK3β in the adult rat retina and the effects of siGSK3β on RGC survival, axon regeneration, and glial activation after ONC in vivo. In the intact uninjured retina, GSK3β was primarily localised to Brn3a^+^ RGCs in the GCL (Figure 7A), results similar to those obtained from retinal cultures (see Figure 4B). RGC survival was reduced after ONC + PBS or siEGFP, compared to intact controls, reducing numbers of Brn3a^+^ RGCs/250 µm GCL from 20 ± 0.5 to 11 ± 1 (Figure 7B,C). However, after ONC and the intravitreal delivery of siGSK3β, RGC survival was improved significantly to 14 ± 1/250 µm GCL (*p* < 0.01; Figure 7B,C). Thus, suppression of GSK3β promoted modest, but significant, RGC neuroprotection at 24 days after ONC.

Although there was a trend towards greater numbers of GAP43^+^ axons regenerating beyond the ONC site (*) into the distal ON after treatment with siGSK3β, these differences were small and did not reach significance until 800 and 1200 µm past the lesion site (Figure 7D,E). These results are consistent with the in vitro observations and suggest that, in isolation, GSK3β knockdown promotes limited, but significant, RGC axogenesis.

### 3.5. Reactive Retinal GPAP^+^ Gliosis after ONC + siGSK3β

There was a significant increase in the number of GFAP^+^ Müller glial processes in the IPL after ONC, with 10 ± 2 processes/250 µm IPL (Figure 7F,G). Also, there was no significant difference in Müller glial activation between eyes injected with either PBS, siEGFP, or siGSK3β (Figure 7F,G). These results indicated that GSK3β signalling did not affect Muller glial activation in vivo.

### 3.6. Effects of Combined siGSK3β + siRTP801 on Target Knockdown, RGC Survival, and Neurite Outgrowth

We then determined if a combination of siGSK3β + siRTP801 enhanced RGC survival and neurite/axon regeneration in vitro and in vivo, respectively. After titration of the combined siRNAs, 25 nM each of siGSK3β and siRTP801 optimally and significantly reduced RTP801 and GSK3β mRNA (Figure 8A) and immunoreactivity compared to sNBA or siEGFP treated cultures (Figure 8B). Combined siGSK3β + siRTP801 significantly knocked down the number of RGCs expressing both GSK3β and RTP801^+^ by 40% (*p* < 0.0001; Figure 8C,D), significantly improved RGC survival by 50% (*p* < 0.001; Figure 8E), increased the number of RGCs bearing neurites from 21 ± 2% to 38 ± 3% (*p* < 0.05; Figure 8F), increased the mean longest neurite length from 30.8 ± 3.7 µm to 80 ± 4 µm (*p* < 0.001; Figure 8G), when compared to siEGFP controls. These results suggest that combined siGSK3β + siRTP801 do not affect RGC survival nor the % of RGC bearing neurites over that induced by the individual siRNAs, but have significant additive effects on the length of longest RGC neurites when combined (compare with Figure 5).

### 3.7. RGC Survival and Axon Regeneration 24 days after ONC and Combined Treatment with siGSK3β and siRTP801

After the injection of combined siGSK3β + siRTP801, RGC survival was significantly greater than after injection of either PBS or siEGFP (*p* < 0.0001), but not significantly different from either siGSK3β or siRTP801 administered singularly (Figure 9A,B). Interestingly, siRTP801 alone promoted significantly more RGC survival than siGSK3β alone (*p* < 0.01; Figure 9A,B). In contrast to single administrations of siRTP801 and siGSK3β, combined delivery of siRTP801 + siGSK3β resulted in a significant increase in the number of GAP43^+^ regenerating axons compared to PBS, siEGFP, and siGSK3β treatments at all distances (Figure 9C,D). There were also more GAP43^+^ regenerating RGC axons in siRTP801 + siGSKβ-treated animals than after siRTP801 treatment alone at each of the distances along the ON [21], although the differences only reached statistical significance at 800 µm. These results demonstrated that the combination of siGSK3β + siRTP801 is superior in promoting long distance RGC axon regeneration after ONC than single administration of either siRNA.

## 4. Discussion

Here, we have shown in vitro that siRNA-mediated suppression of GSK3β caused a significant neuroprotection of RGCs and enhanced the number of RGCs with neurites, but not their neurite length, in the presence of activated GFAP^+^ retinal glia. Further, that this neuroprotection was dependent (Rapamycin-sensitive), but neurite outgrowth was independent (Rapamycin-insensitive) of mTORC1 activity. In the presence of an inhibitory Nogo-P4 peptide, siGSK3β treatment disinhibited RGC neurite outgrowth to enhance neurite initiation and extension. Combined siGSK3β + siRTP801 treatment was also RGC neuroprotective and promoted non-additive/non-synergistic neurite outgrowth and elongation in the presence of activated GFAP^+^ retinal glia in culture. After ONC in vivo, siGSK3β was RGC neuroprotective and modestly stimulated numbers of regenerating RGC axons in the ON beyond that observed for PBS and siEGFP-treated controls. However, combined siGSK3β + siRTP801 treatment better neuroprotected RGCs and increased both the number and length of regenerating axons over that observed for individual treatments with siGSK3β or siRTP801 [21].

### 4.1. Role of Activated GFAP^+^ Retinal Glia in GSK3β-Mediated RGC Survival and Axon Regeneration

In the in vivo experiments, GFAP^+^ retinal astrocytes and Müller cells were activated after ONC and probably produced growth factors claimed to promote the survival of axotomised RGCs and the regeneration of their axons [26,28,29,30]. To simulate the in vivo retinal ONC paradigm, we prepared activated retinal glial cultures to evaluate the contribution of activated retinal glial-derived factors to the siGSK3β effects. We initially prepared non-activated glial cultures and performed all of the in vitro experiments described here and this showed that there was a subpopulation of RGCs (probably intrinsically photosensitive (ip) RGCs; see later) that survived irrespective of either the presence or absence of GFAP^+^ glia. Although siGSK3β treatment in vitro significantly increased RGC survival and the number of RGCs with neurites without affecting their length compared to controls, this was not different in either the presence or absence of activated retinal glia.

Moreover, despite Rapamycin treatment reducing the number of pS6^+^ RGCs, this had no effect on neurite outgrowth parameters, but did reduce RGC survival. Thus, we have been unable to obtain any in vitro evidence for a role of GSK3β in RGC neurite elongation through either activated GFAP^+^ retinal glia or by activating the mTOR signalling pathway, and conclude that the endogenous activated glial reaction seen in vivo after ONC was irrelevant to GSK3β-stimulated RGC axon regeneration. GSK3β was expressed in RGCs but not in activated retinal glia, and thus the neuroprotective effects we see after GSK3β treatment are explained by specific RGC targeting of siGSK3β.

### 4.2. GSK3β and Neuroprotection

Although a role for GSK3β in apoptosis is controversial [31,32,33,34,35], dual-regulation has been proposed, promoting intrinsic but inhibiting extrinsic apoptotic pathways by transcription of caspase-3 [36], pro-apoptotic p53 tumour suppressor [37], and bax genes [38]. Conversely, transfection with siGSK3β neuroprotects against glutamate-induced NMDA-receptor-mediated toxicity [39,40] and shGSK3β suppresses the up-regulation of pro-apoptotic dynamin-related protein 1 (Drp1) in the retina in response to raised intra-ocular pressure [41]. Lithium-induced GSK3β inhibition protects against the death of axotomised RGCs [42], although lithium enhances mTOR activity downstream of Akt and also affects other kinases and phosphatases. Here, Rapamycin obliterated pS6 expression in RGC and reduced siGSK3β-induced RGC neuroprotection to control levels, suggesting recruitment of an mTORC1-dependent anti-apoptotic mechanism, mediated through pS6 [2,43]. An alternative mTORC1 transcription route through 4E-BP134 could also be active, since RGC survive after ONC in 4E-BP knock-out mice [44], possibly through a direct link (recently demonstrated in a cancer cell line [45]) between GSK3β and 4E-BP1. Most pS6^+^ RGCs are melanopsin intrinsically photosensitive (ip) RGC and are resistant to apoptosis [46], and thus siGSK3β probably mediates neuroprotection of non-ipRGCs at least in part by activation of either 4E-BP or other substrates.

### 4.3. GSK3β and Axon Regeneration

Reports on the role of GSK3β in axon regeneration are currently inconsistent and require further elucidation. For example, studies in peripheral nerve regeneration have shown that sustained GSK3 activity using the GSK3α^S21A^/β^S9A^ [GSK3(α/β)^S/A^] double knock-in mice, in which GSK3α/GSK3β cannot be inactivated by AKT-mediated phosphorylation, nerve regeneration was either unaffected [47] or markedly facilitated [9]. In the CNS, pharmacological inactivation of GSK3 by lithium administration stimulated axon regeneration after spinal cord injury [6]. In the CNS however, and in particular in RGCs, GSK3β KO enhanced inflammatory stimulation-mediated and AKT-induced RGC axon regeneration [7,10]. In addition, inactivation of eIF2B-epsilon reduced both GSK3β and AKT-mediated axon regeneration whilst elevated GSK3 activity enhanced RGC neurite outgrowth only when CRMP2 activity was maintained [7,10], suggesting that several downstream effectors are important in GSK3β-mediated RGC axon regeneration.

GSK3β is a key axogenic factor [12,48] regulating *ntf* gene transcription and the phosphorylation of multiple axon growth substrates, including the nuclear factor of activated T cells (NFAT), CREB, and β-catenin [3], many of which may be potentiated by activated retinal glial-derived NTF known to influence gene transcription via CREB [49,50]. For example, GSK3β inhibition after NGF activation of the PI3K-Akt axis is required for axon growth [51], but Akt-induced phosphorylation of GSK3β may not be the sole determinant of GSK3β activity [52]. In our study, GSK3β suppression alone enhanced the number of RGCs bearing neurites (i.e., initiation) and increased the length of the longest neurite (i.e., elongation). This suggests that GSK3β is important for RGC growth initiation and axon elongation. The same results were recapitulated in the in vivo paradigm where more axons were present at all distances beyond the ONC site, but the effect only became significant at 800 and 1200 µm.

In the absence of activated retinal glia-derived NTF in vitro, siRNA suppression of GSK3β baseline activity may initiate axon growth on a permissive laminin substrate. Treatment of retinal cultures with siGSK3β did not increase the frequency of RGCs expressing pGSK3β(Ser9), pCRMP2, pSmad1, and pMAP1B, but alternative substrates include: (i), Tyr216, which potentiates of GSK3β activity either by auto-phosphorylation or post-translational modification [3,53]; (ii), Smad1, which increases pGSK3β(Ser9) and concomitantly decreases pCRMP2, reversing axon growth repression induced by PI3K-inhibitors [48]; and (iii), phosphorylated and non-phosphorylated CRMP2, which promote axonal elongation by regulating microtubule assembly in axonal growth cones, an effect which is reduced by GSK3β-dependent phosphorylation [54].

We found that siGSK3β altered RGC expression of the GSK3β substrate pCRMP2 downstream of RhoA/ROCK, while others have shown that inactivation of CRMP2 by phosphorylation inhibits microtubule polymerisation, resulting in axon/neurite growth collapse [55,56,57]. MAP1B is activated in response to GSK3β-dependent phosphorylation, leading to polymerisation of axonal growth cone microtubules and growth cone advancement [3,15]. However, our unpublished findings indicate that pMAP1B did not play a major role in siGSK3β-induced RGC neurite growth and is consistent with results showing that, in spinal neurons in which GSK3β is deleted, axon growth is also independent of MAP1B activity [12]. Nonetheless, MAP1B may function as a GSK3β substrate in initial axonal polarisation and growth during development [58].

GSK3β regulates microtubule dynamics in axon growth cones and is thus implicated in growth cone collapse induced by CNS myelin and scar-derived inhibitors [58,59]. Our observation that GSK3β suppression disinhibited RGC neurite growth in the presence of Nogo-P4 peptide agrees with observations that myelin associated glycoprotein- and chondroitin sulphate proteoglycan-activated GSK3β signals reduce neuron number and suppress neurite elongation, all of which are reversed by GSK3β inhibitors [16,17]. However, the effect of siGSK3β treatment was marginal and only brought neurite outgrowth to levels observed with CNTF alone. Paradoxically, GSK3β over-expression attenuates myelin-dependent axon growth inhibition through interaction between CRMP4 and RhoA and, conversely, GSK3β inhibitors repress neurite outgrowth and neither have growth promoting nor repressive effects in the presence of CNS axon growth inhibitors [8]. These conflicting observations probably relate to relative activation/inhibition of the diverse range of GSK3β substrates such as APC, CLASP, Tau, CREB [3,60], and their moderation by phosphorylation, cellular localisation (e.g., soma vs. axon), and stages of development [3].

### 4.4. Combined siRNA Approach for RGC Neuroprotection and Axon Regeneration

In vitro, combined treatment with siRTP801 + siGSK3β reduced RGC GSK3β and RTP801 expression by 40%, but only improved neurite elongation and not initiation nor RGC survival compared to each siRNA given separately (for separate delivery of siRTP801, see [21]). In vivo, the RGC neuroprotection afforded by siRTP801 + siGSK3β combination was neither additive nor synergistic compared with siRTP801 and siGSK3β given alone, although the combination promoted significantly more GAP43^+^ RGC axons to regenerate in the ON for longer distances (i.e., axon elongation) than in PBS and siEGFP controls or after siGSK3β alone treatment. siGSK3β may protect growth cones from collapse by suppression of growth inhibitory ligands, but treatment with siRTP801, coupled with activated retinal glia-derived NTF, promoted limited RGC axon regeneration; probably explained if residual GSK3β activity acts to subdue the regenerative effect without altering RGC viability, thereby accounting for both the inhibitory action of TSC on mTORC1/2 activity and potentiated RGC axon regeneration by moderating axonal growth cone dynamics after suppression of GSK3β activity. We therefore propose a working hypothesis that RGC survival is mediated through mTOR activation, whilst RGC axon regeneration is not. Hence, RGC survival is not additive after GSK3β and RTP801 knockdown, but RGC axon regeneration is enhanced by the combined treatment, probably independently of each other (Figure 10).

## 5. Conclusions

In conclusion, GSK3β promotes RGC death possibly by suppressing downstream mTORC1 activity, while siGSK3β knockdown enhances RGC viability, possibly by activating the mTORC1-pS6/4E-BP1 axis; effects that were robust in vitro but more muted in vivo where the presence of multiple other signalling complexes, including ON scar-derived, and possible activated retinal glial-factors, probably also contribute. In vitro, siGSK3β did not affect neurite length, but promoted initiation of RGC neurite outgrowth, activities shown to be independent of activated retinal glia. In vivo, siGSK3β treatment alone also enhanced RGC survival and modest axon growth initiation in the transected ON. Furthermore, RGC survival was no greater after combined treatment with siRTP801 + siGSK3β than with either siRTP801 or siGSK3β alone, but both axon initiation and elongation were significantly enhanced by the combined treatment.

## Figures and Tables

**Figure 1 cells-08-00956-f001:**
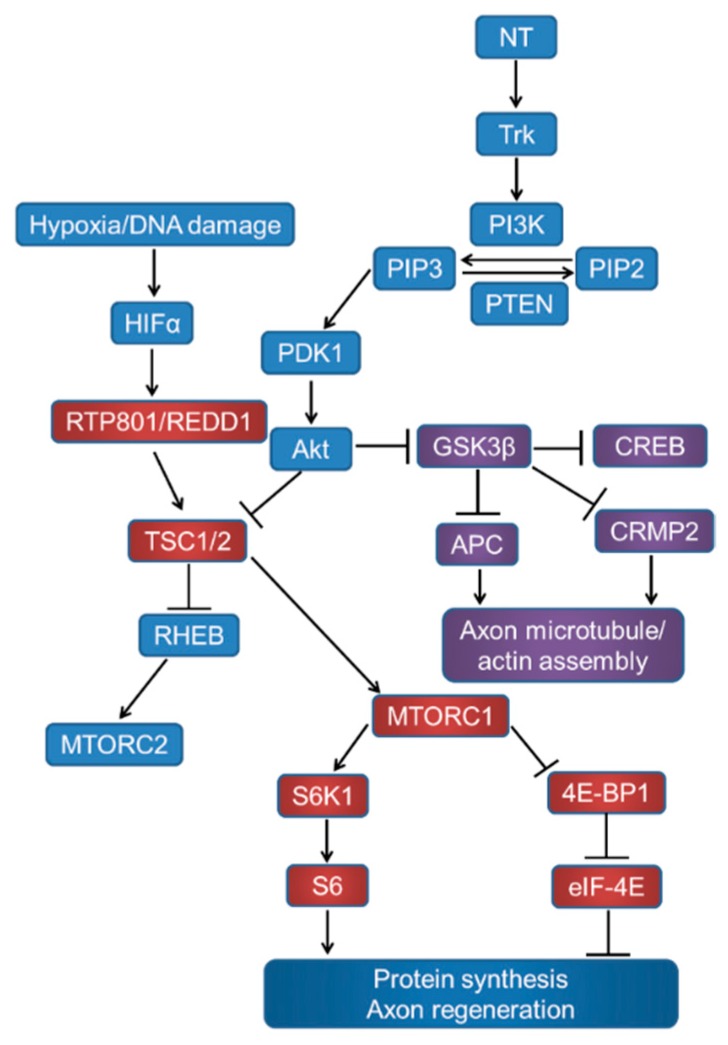
Signalling through the mTORC1, GSK3β, and pathways related to proteins synthesis and axon regeneration. Neurotrophins (NT) act on tyrosine kinase (Trk) receptors and induce phosphatidylinositol kinase (PI3K) activity, which converts phosphatidylinositol (4,5) bisphosphate (PIP2) to phosphatidylinositol (3,4,5) triphosphate (PIP3). Phosphatase and tensin homolog deleted on chromosome 10 (PTEN) catalyses the reverse reaction. PIP3 activates phosphatidylinositol-dependent protein kinase 1 (PDK1), Akt phosphorylation, and inhibits tuberous sclerosis complex (TSC1/2). TSC1/2 can stimulate the Ras homolog enriched in the brain (Rheb) to upregulate mTOR activity. Akt can also inhibit GSK3β, which in turn disinhibits CREB-mediated NT transcription, APC, and CRMP2 to promote growth cone assembly.

**Figure 2 cells-08-00956-f002:**
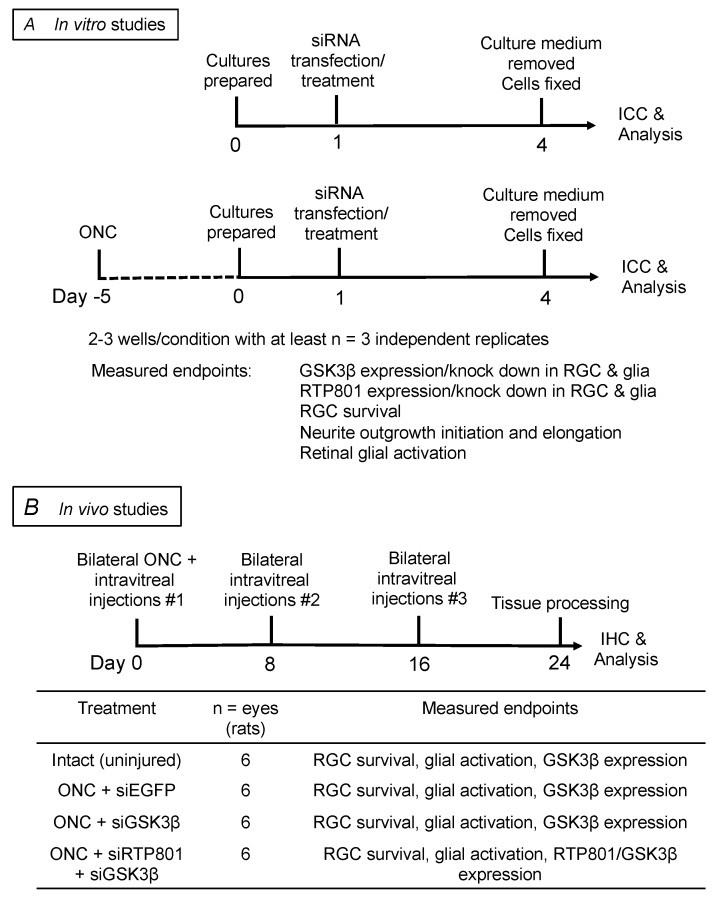
Schematic of the in vitro and in vivo studies, treatment paradigms, and their timelines. (**A**) experimental plan for in vitro studies; (**B**) experimental plan for in vivo studies.

**Figure 3 cells-08-00956-f003:**
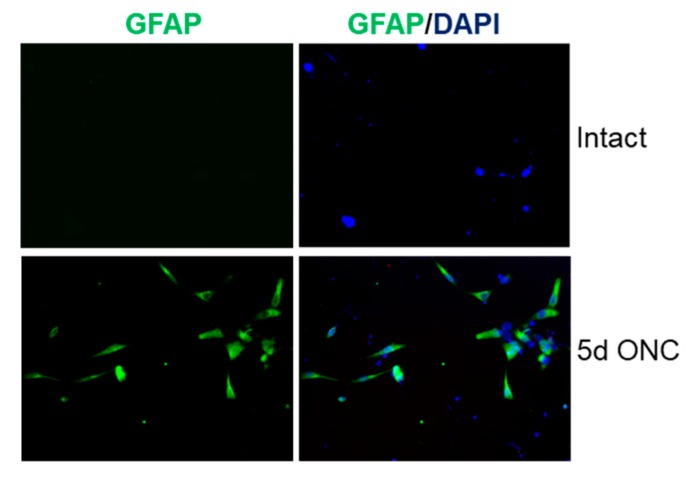
GFAP immunoreactivity in retinal cultures prepared from intact adult rats and at 5 days after ONC. Few, if any, GFAP^+^ glia were present in cultures from intact rats whereas lots of GFAP^+^ glia were present in cultures prepared from rats at five days after ONC.

**Figure 4 cells-08-00956-f004:**
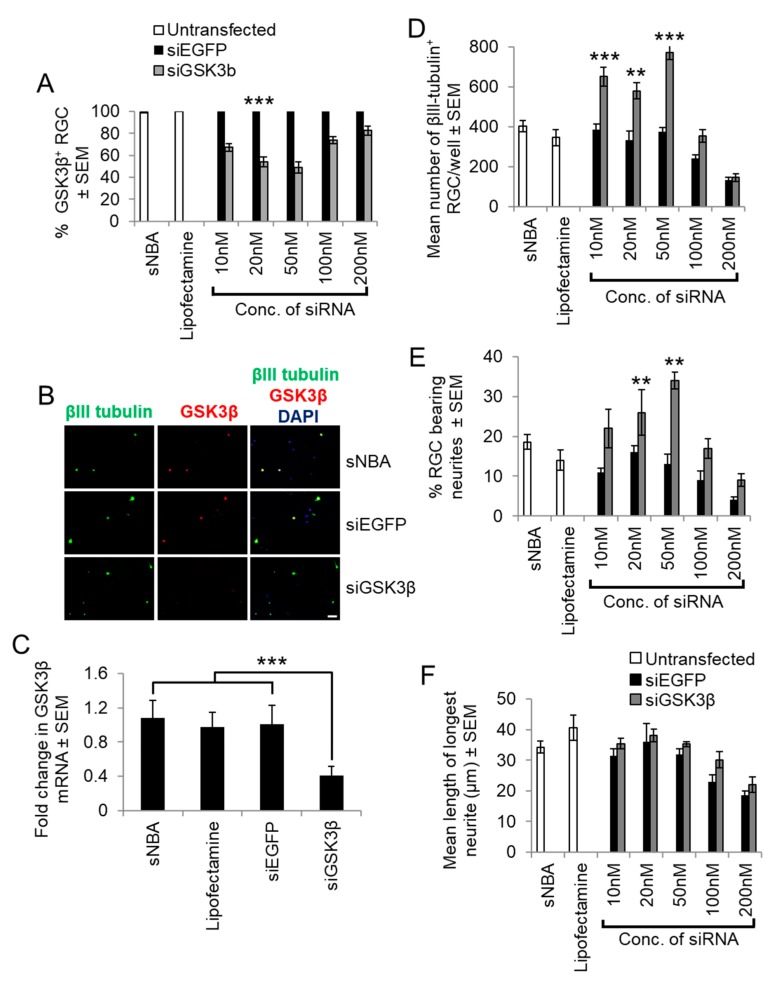
siGSK3β-mediated knockdown in retinal cultures prepared from adult rats at five days after ONC. (**A**) Quantification of the proportion of GSK3β^+^ RGC with increasing concentrations of siGSK3β when compared to sNBA and Lipofectamine 2000-treated controls. (**B**) Representative retinal cultures established five days after ONC in vivo to activate retinal glia and treated with siGSK3β (50 nM) demonstrated a lack of GSK3β (red) detection in βIII-tubulin^+^ RGC (green), whilst abundant immunoreactivity was present in βIII-tubulin^+^ RGC in sNBA and siEGFP-treated (50 nM) control cultures. (Scale bar in B = 20 µm; ** = *p* < 0.01; *** = *p* < 0.001). (**C**) Analysis of GSK3β mRNA levels in cultured adult rat retinal cells after transfection with the optimal concentration of siGSK3β (50 nM) to confirm GSK3β knockdown. (*** = *p* < 0.001). (**D**) Quantification of the proportion of surviving βIII-tubulin^+^ RGC with increasing concentrations of siGSK3β shows that 50 nM optimally promotes RGC survival. (**E**) Quantification of the proportion of RGC bearing neurites with increasing concentrations of siGSK3β shows that 50 nM was optimal to stimulate initiation of neurite outgrowth. (**F**) Quantification of the longest RGC neurite length shows that increasing concentrations of siGSK3β does not affect the length. (** = *p* < 0.01; *** = *p* < 0.001). *n* = 2 wells/treatment, 3 independent repeats (total *n* = 6 wells/treatment).

**Figure 5 cells-08-00956-f005:**
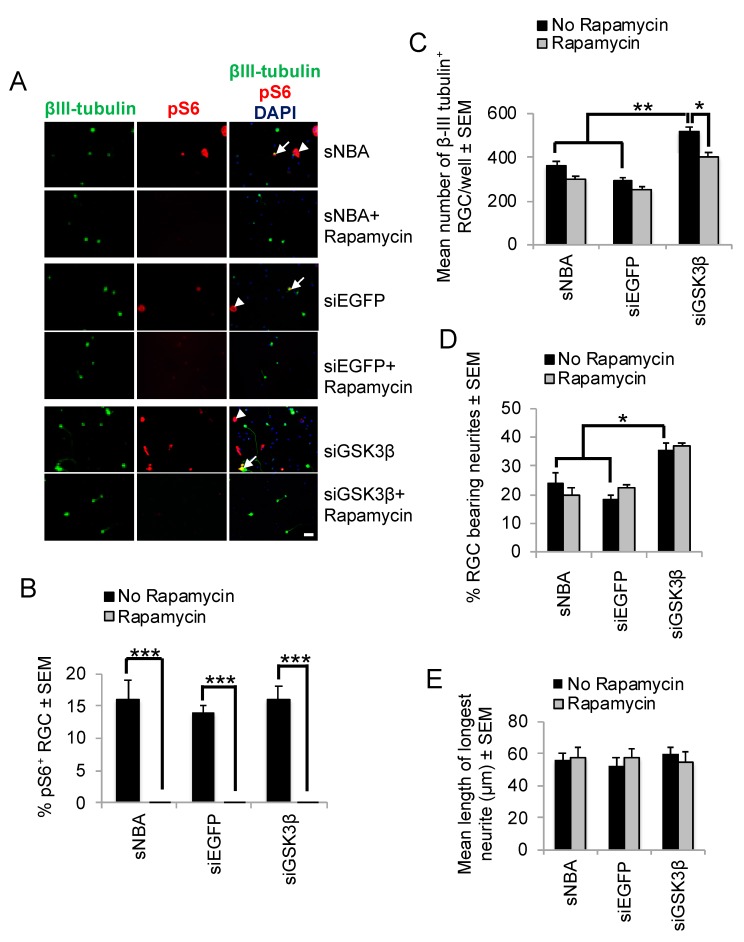
Effects of Rapamycin on siGSK3β-mediated RGC survival and mTORC1 activity in vitro. (**A**) Representative retinal cultures established five days after ONC in vivo to activate retinal glia, treated for three days with sNBA, siEGFP, and siGSK3β in the presence and absence of the mTORC1 inhibitor Rapamycin and immunostained for βIII-tubulin (green) and pS6 (red), with DAPI (blue) as a nuclear counterstain. Note that pS6 is detected both in βIII-tubulin^+^ RGC (long arrows) and βIII-tubulin^−^ cells (short arrows) and the abolition of pS6 immunostaining in the presence of Rapamycin. (**B**) The % of RGC exhibiting pS6 immunoreactivity indicated abolition of pS6 expression in the presence of Rapamycin, but there was no significant effect of siGSK3β on the proportion of pS6^+^ RGC (* = *p* < 0.05, ** = *p* < 0.01, *** = *p* < 0.001). (**C**) Quantification of the number surviving βIII-tubulin^+^ RGC after treatment with siGSK3β in the presence of Rapamycin shows that RGC survival is significantly enhanced. (Scale bar in A = 20 μm; ** = *p* < 0.01). (**D**) Quantification of the % RGC bearing neurites and (**E**) length of the longest RGC neurite are both significantly enhanced after siGSK3β treatment, but this effect was unaffected by Rapamycin. *n* = 2 wells/treatment, 3 independent repeats (total *n* = 6 wells/treatment).

**Figure 6 cells-08-00956-f006:**
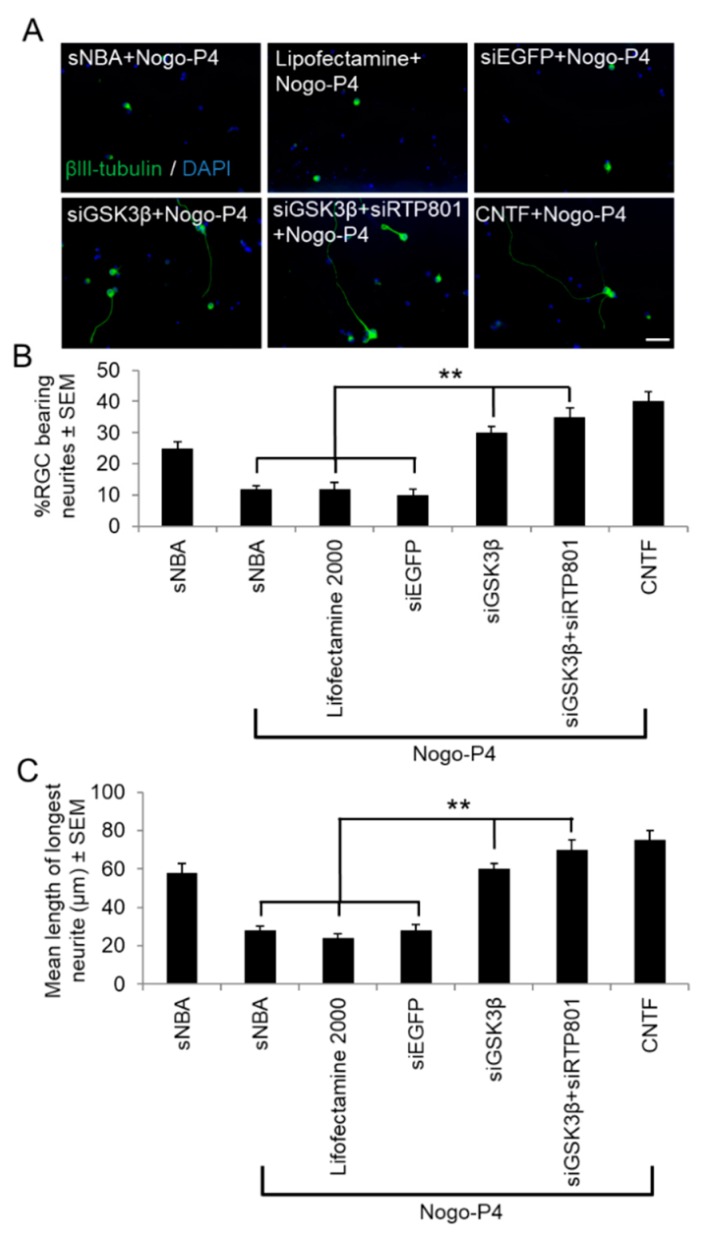
Inhibition of GSK3β activity promotes disinhibition of RGC neurite outgrowth in the presence of inhibitory Nogo-P4 peptide. (**A**) Immunocytochemistry showing disinhibited neurite outgrowth from βIII-tubulin^+^ RGC in the presence of Nogo-P4 peptide after treatment with siGSK3β whilst control cultures treated with sNBA, Lipofectamine 2000, or siEGFP were unable to achieve disinhibited neurite outgrowth. CNTF-treated cultures were used as positive controls. (**B**) Quantification of the proportion of RGC bearing neurites and (**C**) the mean longest RGC neurite after siRNA treatment in the presence of Nogo-P4 peptide, showing significant disinhibition of neurite outgrowth after treatment with siGSK3β. (Scale bar in A = 20 μm; ** = *p* < 0.01). *n* = 2 wells/treatment, 3 independent repeats (total *n* = 6 wells/treatment).

**Figure 7 cells-08-00956-f007:**
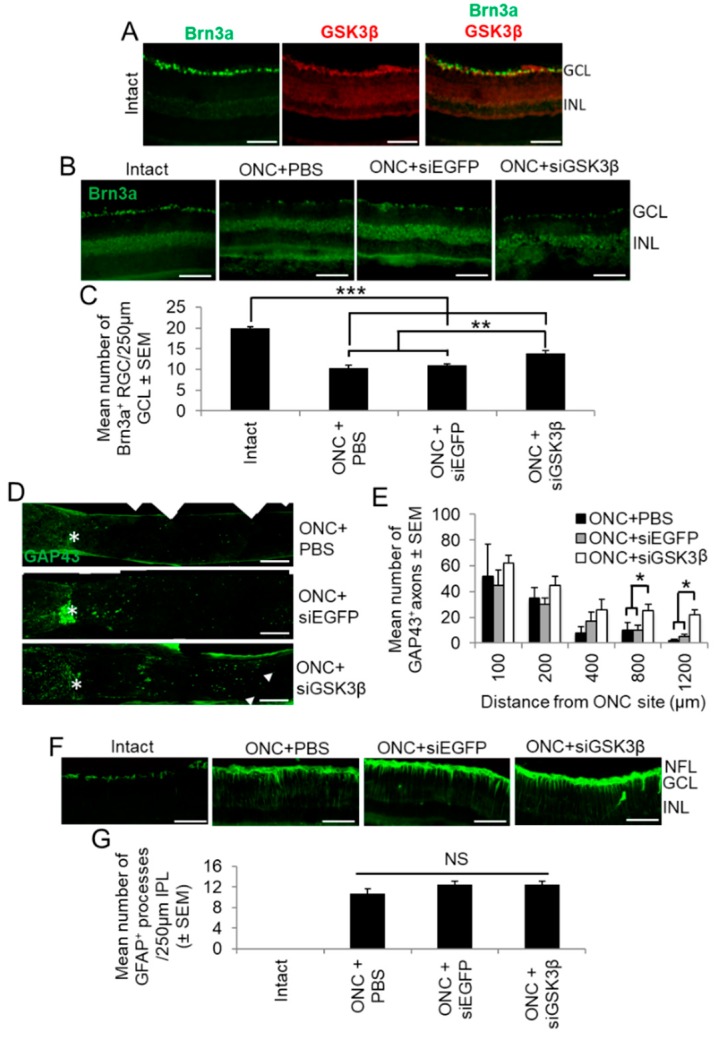
Cellular localisation of GSK3β and the effects of siGSK3β on RGC neuroprotection in vivo. (**A**) Retinal sections from uninjured adult rats immunostained for Brn3a (green) and GSK3β (red), demonstrating that constitutive expression of GSK3β was limited to the GCL, where it localised to Brn3a^+^ RGC (INL = inner nuclear layer). (**B**) siGSK3β protects RGC (Brn3a^+^) from death at 24 days after ONC, compared to ONC + PBS and ONC + siEGFP. Intact uninjured controls show baseline levels of Brn3a^+^ RGC. (**C**) Quantification of Brn3a^+^ RGC survival in the 250 µm counting area of the GCL. (Scale bars in A and B = 100 µm; ** = *p* < 0.01; *** = *p* < 0.001). (**D**) Longitudinal ON sections immunostained to demonstrate GAP43^+^ regenerating axons (arrows) after ONC + PBS (top panel), ONC + siEGFP (middle panel) and ONC + siGSK3β (bottom panel). The asterisk demarcates the ON site and the boxed area in the lower panel represents the magnified area. (**E**) Quantification of GAP43^+^ regenerating axons 100, 200, 400, 800, and 1200 μm beyond the ONC site in eyes after intravitreal injection of PBS, siEGFP, and siGSK3β. (Scale bars = 200 μm; * *p* < 0.05). (**F**) GFAP^+^ glial activation occurs after ONC and is not further enhanced by siGSK3β treatment. (**G**) Quantification of the number of GFAP^+^ glial processes crossing a 250 µm line in the inner plexiform layer corroborates this observation. (Scale bar in F = 100 µm; NS = not significant). *n* = 6 eyes (from 6 rats)/treatment.

**Figure 8 cells-08-00956-f008:**
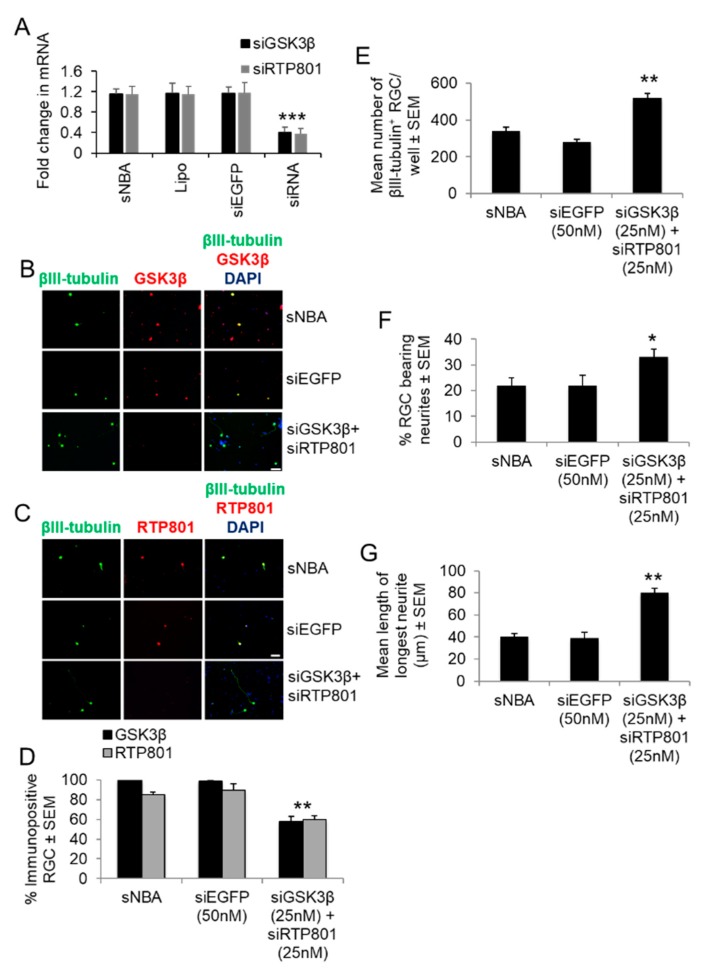
The effect of combined siGSK3β + siRTP801 on GSK3β and RTP801 immunoreactivity in RGC in vitro. (**A**) GSK3β and RTP801 mRNA levels in cultured adult rat retinal cells after transfection with the optimal concentrations of siGSK3β (25 nM each) to confirm significant GSK3β and RTP801 knockdown. Lipo = Lipofectamine 2000 (*** = *p* < 0.001). Both (**B**) RTP801 and (**C**) GSK3β^+^ immunoreactivity was present in RGC from sNBA and siEGFP control cultures, whilst treatment with combined siGSK3β + siRTP801 abolished GSK3β and RTP801 immunoreactivity in RGC. (Scale bar in B and C = 20 μm). Combined siGSK3β + siRTP801 significantly increased (**D**) the number of surviving βIII-tubulin^+^ RGC, (**E**) % of RGC with neurites and (**F**) the mean longest neurite length. (**G**) GSK3β^+^ and RTP801^+^ immunoreactivity was present in sNBA and siEGFP control cultures, whilst treatment with combined siGSK3β + siRTP801 significantly reduced RTP801 and GSK3β immunoreactivity in RGC. * = *p* < 0.05; ***p* < 0.01; *** = *p* < 0.001). *n* = 2 wells/treatment, 3 independent repeats (total *n* = 6 wells/treatment).

**Figure 9 cells-08-00956-f009:**
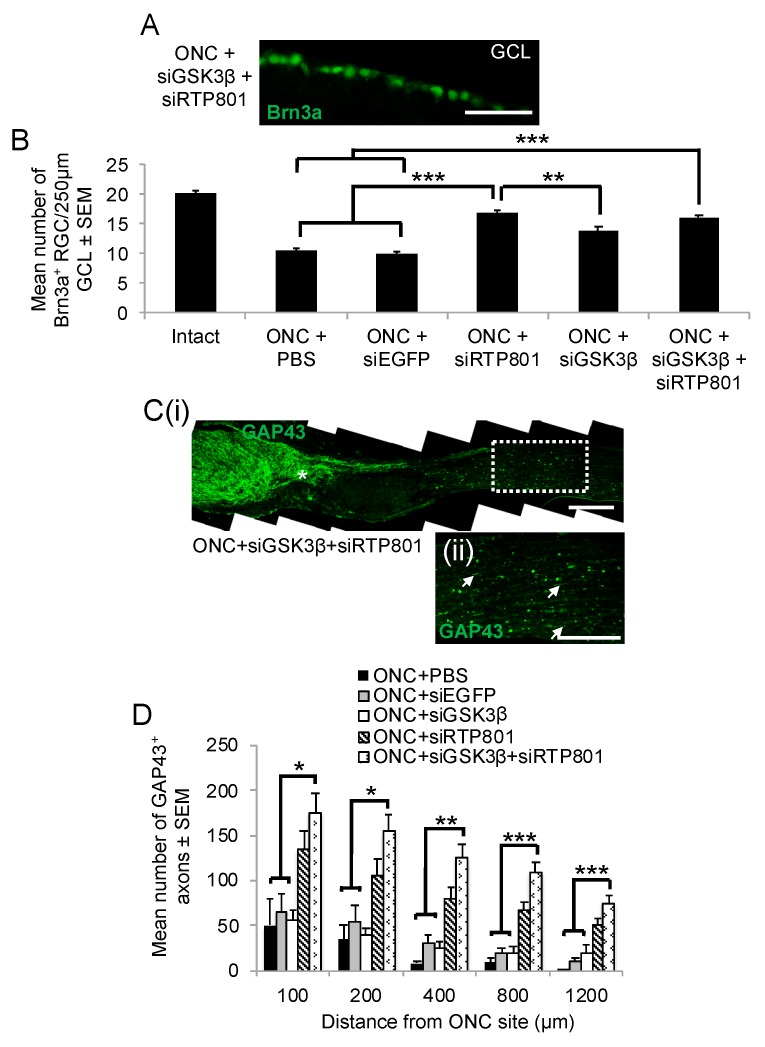
The effect of combined siGSK3β + siRTP801 on RGC survival and axon regeneration 24 days after ONC. Although (**A**) many Brn3a^+^ RGC were present after combined siGSK3β + siRTP801 treatment, (**B**) quantification of the number of RGC/250 µm GCL did not show any additive or synergistic effects of both siRNAs, despite individual and combination treatments being equally RGC neuroprotective. (**C**(**i**)) Many GAP43^+^ regenerating axons were present after combined siGSK3β + siRTP801 treatment; (**C**(**ii**)) shows a high power view of boxed region in **C**(**i**)). (**D**) Quantification of the number of GAP43^+^ axons at all measured distances beyond the lesion site showed that combined siGSK3β + siRTP801 treatment was significantly more axogenic than PBS, siEGFP, or siGSK3β alone treatment. Note that combined siGSK3β + siRTP801 was better than siRTP801 alone treatment, but the difference only became statistically significant at 800 µm. (Scale bars in A and C = 100 µm; * = *p* < 0.05, ** = *p* < 0.01; *** = *p* < 0.001). *n* = 6 eyes (from 6 rats)/treatment.

**Figure 10 cells-08-00956-f010:**
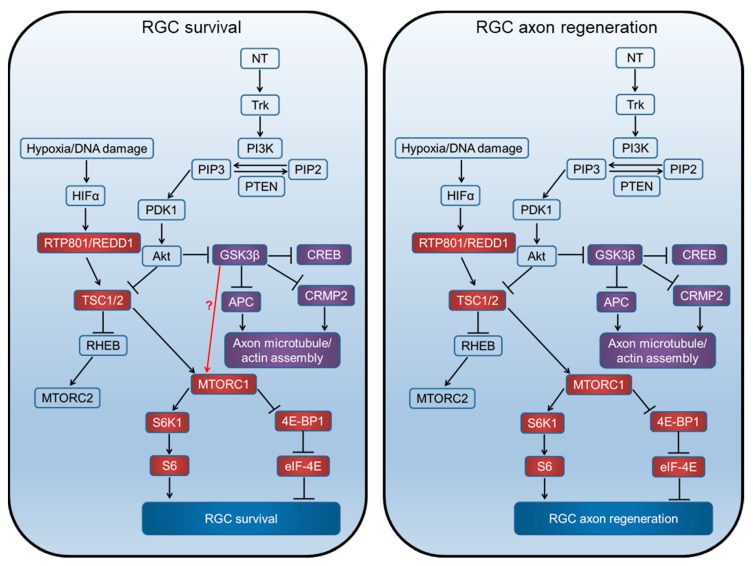
Proposed mechanism of RGC survival and axon regeneration by GSK3β and RTP801. RGC survival is signalled through the mTORC1 pathway downstream of GSK3β, whilst axon regeneration is signalled independently by mTOR and GSK3β pathways. The question mark and the red arrow represent unknown effectors downstream of GSK3β.

**Table 1 cells-08-00956-t001:** List of primary and secondary antibodies used in this study.

	Antigen	Species	Dilution	Supplier
**Primary**	Brn3a	Goat	1:250	Santa Cruz Biotechnology, Santa Cruz, CA, USA
	GAP43	Mouse	1:500	Invitrogen. Paisley, UK
	βIII-tubulin	Mouse	1:200	Sigma, Poole, UK
	GFAP	Mouse	1:200 (ICC),1:250 (IHC)	Sigma, Poole, UK
	Phospho-S6	Rabbit	1:200 (ICC)	Cell Signalling Technology, Hitchin, UK
	RTP801	Rabbit	1:100 (ICC & IHC)	Abcam, Cambridge, UK
	GSK3β	Mouse	1:200 (ICC & IHC)	Abcam, Cambridge, UK
**Secondary**	Goat IgG AlexaFluor 488	Donkey	1:500 (IHC)	Invitrogen. Paisley, UK
	Mouse IgG AlexaFluor 488	Goat	1:400 (ICC), 1:500 (IHC)	Invitrogen. Paisley, UK
	Rabbit IgG AlexaFluor 594	Goat	1:400 (ICC), 1:500 (IHC)	Invitrogen. Paisley, UK

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
