# Peer review of "Effects of siRNA-Mediated Knockdown of GSK3β on Retinal Ganglion Cell Survival and Neurite/Axon Growth"

_cells, 2019, doi:10.3390/cells8090956_

Round 1
Reviewer 1 Report
The manuscript by Ahmed et al. aims to clarify the importance of GSK3B for axonal regeneration following optic nerve crush. The authors use both in vitro and in vivo approaches to demonstrate that knockdown of GSK3B promotes cell survival and axonal regeneration. Furthermore, they show that simultaneous knockdown of both GSK3B and RTP801 can be used to enhance regenerative potential. This is an interesting manuscript presenting potentially important data, however there are areas that need to be addressed prior to considering it further.
1. Further exploration of the relationship between GSK3B and RTP801 would greatly improve the manuscript. This is especially true when the major aim is to help clarify the importance of GSK3B, but only siRNA is used. Do the authors suggest they are functioning in the same pathway? Does the knockdown of one affect the localisation of the other? Can the overexpression of one rescue knockdown of the other, ie is one functioning downstream from the other?
2. The rationale for each set of experiments should be clearly outlined. For the majority of experiments, no rationale is provided leaving the reader to try and work out why they were done and how they fit within the larger narrative. This also made it more difficult to review the manuscript.
3. All of the images provided in the figures are far too small. This made it near impossible to accurately assess what was being shown in each panel.
4. A figure outlining the relationship between RTP801, GSK3B, mTORC1/2 and TSC would be helpful. Also, a summary figure outlining the major findings would greatly help the readability.
5. Abstract.
This should be re-written to simplify and for clarity. The start of the abstract and introduction are exactly the same. This needs to be amended.
6. Methods.
“The sequence and chemical modifications to 122 siGSK3β and siRTP801 were proprietary to Quark Pharmaceuticals Inc”. If the sequences are not being provided, why are the authors trying to publish this research? When the major aim of this research was to help clarify opposing results in the current literature, how can this data possibly achieve this aim when there is no way that other groups can try to replicate it?
qPCR experiments appear to only have been performed in duplicate. Triplicate biological samples need to be used to provide confidence in the data.
7. Data no shown.
All data relevant to this study must be included with the manuscript, either in the main figures or as new supplementary figures.
8. Figure 2.
It is stated that increased concentrations of siGSK3B is potentially neurotoxic, yet panels D and E suggest that siEGFP at higher concentrations leads to similar detrimental effects. Thus, this effect does not appear to be specific to GSK3B and instead appears to be a consequence of the siRNA concentration not sequence.
9. Figure 4.
The effect of simultaneously knocking down GSK3B and RTP801 on reversing the inhibition of Nogo should be analysed here.
10. Figure 6.
These experiments need to be performed alongside single knockdown experiments to allow direct comparisons to be made. Inferences to the effectiveness of double knockdown versus single knockdown when the single knockdowns were not performed at the same time are not accurate and should not be made. The values for the controls in Fig 6G appear almost 50% less than those in 3E.
11. Line 469-471: is this a journal guideline statement left in the manuscript?
12. Discussion
Lines 554-559. How can GSK3B be claimed to solely affect initiation when significant differences in the number of the longest axons were seen? Wouldn’t this suggest a role in elongation?
Author Response
“Further exploration of the relationship between GSK3B and RTP801 would greatly improve the manuscript. This is especially true when the major aim is to help clarify the importance of GSK3B, but only siRNA is used.”
These siRNAs are specific and translationally relevant and hence we have not used any other reagents to modulate GSK3B and RTP801. As for the main aim of our manuscript, we wanted to determine the contribution of GSK3B singly and in combination with RTP801 to promote RGC axon regeneration. We have already published data related to RTP801 previously (Morgan-Warren et al., 2016).
“Do the authors suggest they are functioning in the same pathway? Does the knockdown of one affect the localisation of the other? Can the overexpression of one rescue knockdown of the other, ie is one functioning downstream from the other?”
Our results show that for RGC survival, GSK3B knockdown impacts on the mTOR pathway whereas for axon outgrowth, this was independent of mTOR. We have now added a summary diagram as Figure 10 to explain this relationship.
“The rationale for each set of experiments should be clearly outlined. For the majority of experiments, no rationale is provided leaving the reader to try and work out why they were done and how they fit within the larger narrative. This also made it more difficult to review the manuscript.”
We have now provided rationale for each experiment in results section of the manuscript.
“All of the images provided in the figures are far too small. This made it near impossible to accurately assess what was being shown in each panel.”
All of the images have been enlarged to make it clearer.
“A figure outlining the relationship between RTP801, GSK3B, mTORC1/2 and TSC would be helpful. Also, a summary figure outlining the major findings would greatly help the readability.”
We have added a new Figure 1 to show the relationship between RTP801, GSK3B, mTORC1/2 and TSC and a new summary figure, Figure 10, to outlie the major findings of our study.
“Abstract. This should be re-written to simplify and for clarity. The start of the abstract and introduction are exactly the same. This needs to be amended.”
The Abstract has been amended to simplify and clarify the text.
Methods. “The sequence and chemical modifications to 122 siGSK3β and siRTP801 were proprietary to Quark Pharmaceuticals Inc”. If the sequences are not being provided, why are the authors trying to publish this research? When the major aim of this research was to help clarify opposing results in the current literature, how can this data possibly achieve this aim when there is no way that other groups can try to replicate it?”
We have re-written the sentences (Lines 129-132) to show that siEGFP and siGSK3β sequences have been published before. However, the sequence for siRTP801 is undisclosed since Quark Pharmaceuticals are performing clinical trials with this compound and have submitted patents to protect the IP. Having said that, Quark are happy to share the compound with any researcher subject to completing an MTA and so there is no restriction on any group trying to replicate this work.
“qPCR experiments appear to only have been performed in duplicate. Triplicate biological samples need to be used to provide confidence in the data.”
It has now been clarified that experiments were performed in duplicate and repeated on 3 independent occasions with cultures derived from 3 different animals (Lines 178-180). We are sorry for not making this clear in the original version of the manuscript.
“Data not shown. All data relevant to this study must be included with the manuscript, either in the main figures or as new supplementary figures.”
We have now removed any areas that says ‘data not shown’ and have presented the data in the revised version of the manuscript.
“Figure 2. It is stated that increased concentrations of siGSK3B is potentially neurotoxic, yet panels D and E suggest that siEGFP at higher concentrations leads to similar detrimental effects. Thus, this effect does not appear to be specific to GSK3B and instead appears to be a consequence of the siRNA concentration not sequence.”
We have now clarified that high concentrations of siEGFP were also toxic to RGC (Lines 330-333). Concentration-dependent toxicity is a well-known problem when using siRNA and hence has to be carefully titrated.
“Figure 4. The effect of simultaneously knocking down GSK3B and RTP801 on reversing the inhibition of Nogo should be analysed here.”
A new panel to show RGC neurite outgrowth and its quantification has been added to Figure 4 (now Figure 5) to show this.
“Figure 6. These experiments need to be performed alongside single knockdown experiments to allow direct comparisons to be made. Inferences to the effectiveness of double knockdown versus single knockdown when the single knockdowns were not performed at the same time are not accurate and should not be made. The values for the controls in Fig 6G appear almost 50% less than those in 3E.”
Single knockdown of RTP801 in the same system has already been published by us previously (Morgan-Warren, 2016) using the same siRTP801 sequences as used here. The main focus of this paper is the impact of siGSKb and then its combination with siRTP801.
We are sorry for the confusion in Figure 3E and Figure 6G since Figure 6G was drawn incorrectly. We have now checked the data and have replotted the values which shows that Figure 3E and 6G are comparable.
“Line 469-471: is this a journal guideline statement left in the manuscript?”
Sorry. This is a journal guideline statement and it was accidently left in. Now deleted.
“Discussion. Lines 554-559. How can GSK3B be claimed to solely affect initiation when significant differences in the number of the longest axons were seen? Wouldn’t this suggest a role in elongation?”
We are sorry for this oversight. The reviewer is right that GSK3b alone affects initiation and elongation. This has been amended.
Reviewer 2 Report
Referee comments:
In this timely and comprehensive study the Authors have shown in vitro as well as in vivo the effects of siRNA-mediated suppression of GSK3β on neuroprotection of RGCs and axogenesis.
This interesting and novel study provided a plethora of information on the mechanisms involved in RGCs physiology and regeneration that might likely have a significant impact on therapeutic strategies targeting retinal degenerations. The showed data are solid and convincing and supports main conclusions of the manuscript. I found only a few minor points for the Authors.
Minor points:
In Fig.2 panel F: Untranfected instead of Untrasfected
In fig.6 panel F, axis name: +/- SEM instead of SEM
Lines 469 to 472: “This section may be divided by subheadings. It should provide a concise and precise description of the experimental results, their interpretation as well as the experimental conclusions that can be drawn.” Is this sentence intended for the Authors during the manuscript preparation? If yes, it must be obviously deleted.
Line 541: “…phosphorylation, that nerve regeneration was either unaffected [47] or markedly facilitated [9]”. Probably is better to eliminate “…that…” “phosphorylation, that nerve regeneration was either unaffected [47] or markedly facilitated [9]”.
Author Response
“In Fig. 2 panel F: Untranfected instead of Untrasfected”
Amended.
“In fig.6 panel F, axis name: +/- SEM instead of SEM”
Amended.
“Lines 469 to 472: “This section may be divided by subheadings. It should provide a concise and precise description of the experimental results, their interpretation as well as the experimental conclusions that can be drawn.” Is this sentence intended for the Authors during the manuscript preparation? If yes, it must be obviously deleted.”
Sorry, deleted.
“Line 541: “…phosphorylation, that nerve regeneration was either unaffected [47] or markedly facilitated [9]”. Probably is better to eliminate “…that…” “phosphorylation, that nerve regeneration was either unaffected [47] or markedly facilitated [9]”.
“that” deleted.
Round 2
Reviewer 1 Report
The authors have address all of my previous concerns. Rationale is now provided for each set of experiments, which together with the additional schematics greatly help with the readability of the manuscript.
I have no further concerns and now support publication.